# Pressure Fluctuation Characteristics of High-Speed Centrifugal Pump with Enlarged Flow Design

**Jianyi Zhang, Hao Yang \*, Haibing Liu, Liang Xu and Yuwei Lv**

The Province Key Laboratory of Fluid Transmission Technology, Zhejiang Sci-Tech University, Hangzhou 310018, China; zdreamx@zstu.edu.cn (J.Z.); haibing9804@sina.com (H.L.); m15895226845@163.com (L.X.); lvyw0902@163.com (Y.L.)
\* Correspondence: JusticeYh@163.com

**Abstract:** The pressure fluctuations of high-speed centrifugal pumps are the hotspot in pump research. Pressure fluctuations is differ for different structural designs and flow structures. High-speed centrifugal pumps are usually designed to increase efficiency with an enlarged flow design at a low specific speed, which changes the structure of the pump. In order to analyze the pressure fluctuations of a high-speed centrifugal pump with an enlarged flow design, the pressure was measured, and the flow field of the pump was simulated with different flow rates. Through analysis, we found that pressure fluctuations varied periodically and was consistent with the blade frequency. The pressure fluctuations at the guide vane and the interference region were also closely related to the vortices at the impeller outlet, which changed differently at different flow rates. The results showed that the high-speed centrifugal pump with an enlarged design had better performance at a large flow rate. The results in this paper can provide reference for the design of a pump that should be designed with the enlarged flow method.

**Keywords:** high-speed centrifugal pump; enlarged flow design; pressure fluctuations; vortex

## 1. Introduction

The high-speed centrifugal pump is the key equipment in the chemical industry, as it transports working media to every unit of the system. It has allowed for high-power density with the development of the processing industry. Increasing speed is the main method to realizing such high power density. However, increases in the rotating speed can result in the flow field changing dramatically, which can make the pressure stronger. This poses a huge challenge for pump development. Therefore, it is very important to study pressure fluctuations in high-speed centrifugal pumps.

For pressure fluctuations, Bertent et al. [1] introduced the measurement method of pressure fluctuations under different conditions. Luo et al. [2] found that the dynamic and static interference between the volute and the tongue is the main source of pressure fluctuations, and that the amplitude of pressure fluctuations is stronger at a low flow rate. Zhang et al. [3] analyzed the influence of the dynamic and static interference between the impeller and the guide vane on the exciting force. The result showed that the fluid force is the smallest when the number of the impeller blades and guide vane is the same. Si et al. [4] studied the high-speed centrifugal pumps at different speeds, and found that the pressure fluctuations' frequency is the same as the blade passing frequency. Yuan et al. [5] conducted research on the internal flow field characteristics of high-speed centrifugal pumps under the condition of vortex. The result showed that the vortex frequency has a great influence on the number of vortices under the non-standard conditions, which affects the pump pressure fluctuations. Li et al. [6] found that the fluctuations become stronger in amplitude when cavitation occurs. Liu's [7] simulation showed that the pump with splitter blades enables pressure changes in a relatively decreased gradient, and the cross-section of static pressure displays better uniformity. Ye et al. [8] pointed out that the splitter blades weaken

the pressure fluctuations of the pump noticeably, and the pressure fluctuations are closely related to the number of blades. Zhu et al. [9] found that the relative position between the guide vane and the tongue has a great influence on the pressure fluctuations' coefficient. The vibration of the pump is closely related to the pressure fluctuations. The pressure fluctuations have an effect on the exciting force, thus affecting the vibration of the pump. For the exciting force, Moore et al. [10] obtained the exciting force on the impellers in two vertical directions, and its accuracy was verified through experiments. Dou et al. [11] simulated the flow field of the pump. It was found that the exciting force can be divided into two parts: the constant force and the pulsating force. The frequency and harmonic frequency of the pulsating force appear in the pump rotor. González et al. [12] found that there is a certain relationship between the impeller diameter and the radial component of the exciting force.

The enlarged flow design method can be used to improve the efficiency of a pump at a low specific speed. The structure of a pump designed by the enlarged flow method is different from the pump mentioned above, which has an impact on the flow field. However, there is little research on the pressure fluctuations of high-speed centrifugal pumps with an enlarged flow design. This is the first study on the pressure pulsation of high-speed centrifugal pumps with an enlarged flow design and its cause, and the differences with different flow rates. This can provide references for pump structure design, pressure pulsation suppression, operating conditions design, and fault diagnosis with the enlarged flow design.

## 2. Theory of Enlarged Flow Design

The enlarged flow design method is expressed as a formula as follows:

$$Q'_v = k_1 q_v, \ n'_s = k_2 n_s \tag{1}$$

where $q_v$ and $n_s$ are the working flow rate and the specific speed of the pump, respectively; $Q'_v$ and $n'_s$ are the flow and specific speed of the pump after using the enlarged flow design method, respectively; and $k_1$ and $k_2$ are the amplification coefficients of the flow rate and specific speed, respectively.

The specific rotational speed becomes amplified with an increase in the flow rate. The performance of the pump drops when the specific speed exceeds a certain value. Therefore, the amplification coefficients of the enlarged flow design method must be selected reasonably. Generally, the amplification coefficients are selected according to the traditional empirical method, which are not reasonable in application. Due to the insufficiency of the traditional empirical method, Yang [13] proposed a method for calculating the reasonable amplification factor of the flow and specific speed of a centrifugal pump based on performance prediction. In order to improve the calculation accuracy and to obtain the most accurate amplification factor, Yang introduced an equation in the general expression of the enlarged flow design method:

$$H' = k_3 H, \ k_2 = \frac{k_1^{1/2}}{k_3^{3/4}} \tag{2}$$

where $H$ is the head of the pump, $H'$ is the head of the pump after using the enlarged flow design method, and $k_3$ is the amplification coefficient of the head. In the design of the pump structure, the exit position angle $\beta_2$ and impeller diameter $D_2$ remain unchanged. The impeller must be amended to ensure that the head remains unchanged. The changes are as follows:

$$b'_2 = b_2 k_2^{5/6} k_1^{1/3}, \ R'_2 = R_2 + dR_2 \tag{3}$$

$$d(H_t - h_s) = 0 \tag{4}$$

where $b_2$ and $R_2$ are the impeller outlet width and outer radius, respectively; $b'_2$ and $R'_2$ are the impeller outlet width and impeller outer radius after using the enlarged flow design

method, respectively; $H_t$ is the theoretical head of the pump; $h_s$ is the total loss in the pump; and $dR_2$ can be obtained according to Equation (4).

The amplification factor and parameters of the pump structure can be obtained through the above formula. The maximum efficiency is obtained after adjustments and repeated calculations.

## 3. Experiment

The experimental data of the pressure fluctuations came from the bench. The bench was composed of a pump body, control table, water tank, pipeline system, sensor, and a data acquisition and analysis system, as shown in Figure 1. The data of pressure fluctuations and vibration were measured under different working conditions by adjusting the valve and electromagnetic flowmeter, which were collected by a data acquisition device.

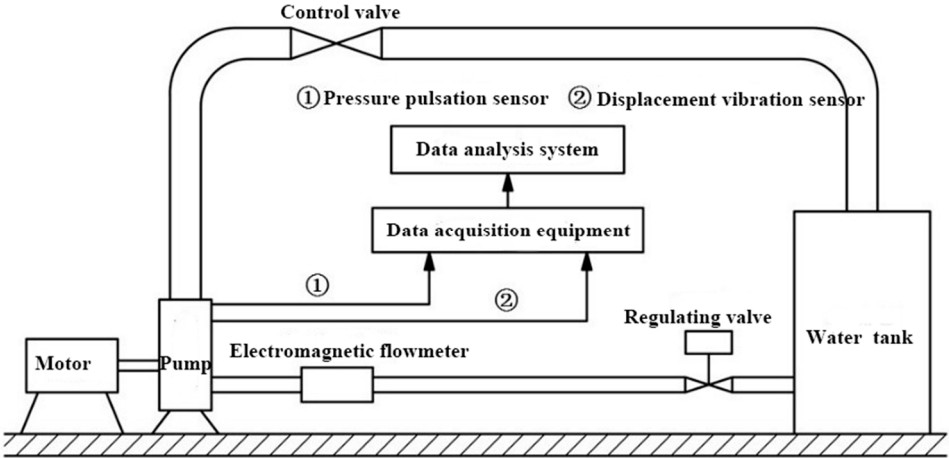

**Figure 1.** Figure of circulation circuit of the pump.

The high-speed centrifugal pump and the position of the sensor are shown in Figure 2. The sensitivity of the pressure fluctuations sensor was 1000 MV/MPa, and the measuring range was 0–5 MPa. The vibration sensor was an eddy current displacement sensor. Three circumferential uniform pressure fluctuations sensors were installed in the flow field of the impeller and the guide vane, and three circumferential sensors were installed in the trailing edge of the inducer.

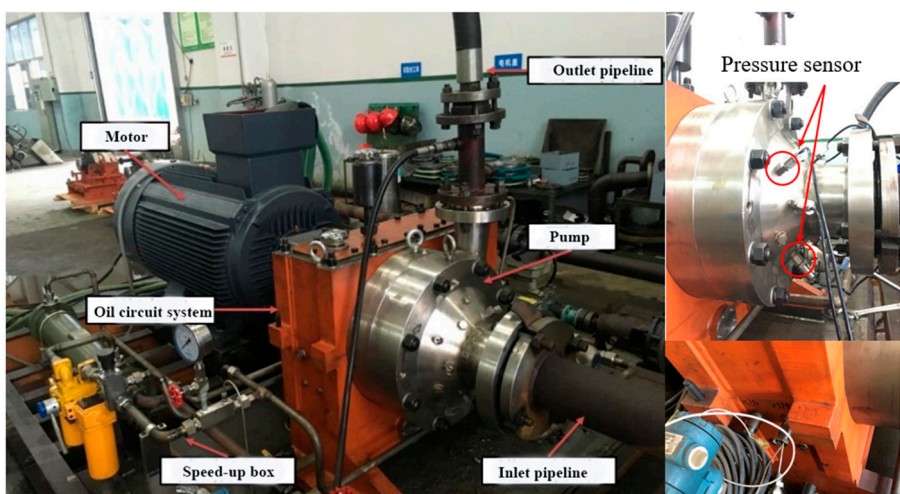

**Figure 2.** Figure of the pump.

The data for the pressure fluctuations and the corresponding frequency under the rated operating condition of 130 m$^3$/h are shown in Figure 3. The pressure ranges from 2.0

to 2.8 MPa. As can be seen in the figure, the main frequency of the pressure is between 900 and 1000 Hz, and the secondary frequency is between 1900 and 2000 Hz.

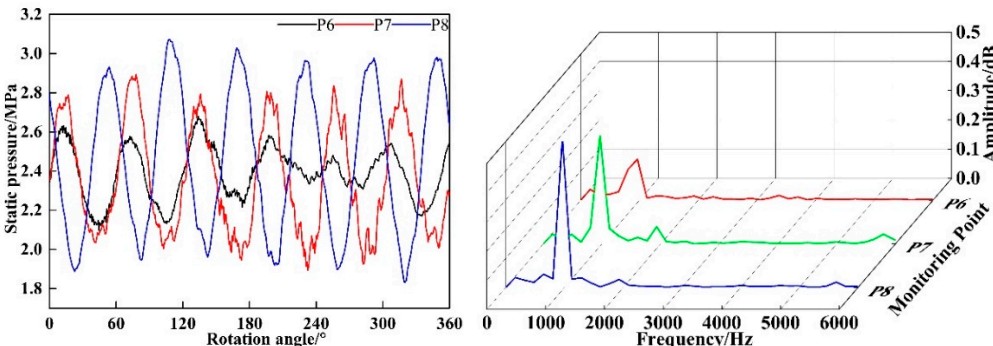

**Figure 3.** The data for pressure fluctuations and frequency domain at Q = 130 m$^3$/h.

The head and efficiency of the pump under different flow conditions were obtained through calculation. In this study, 20 m$^3$/h was used as the interval, and the flow rates were 10, 30, 50, 70, 90, 110, 130, and 150 m$^3$/h. As can be seen in Figure 4, the flow–head-efficiency curves obtained from the numerical simulations and the experiments are nearly identical, and the two curves of the same type show little difference. The simulation results are slightly higher than the experimental values. This is due to some of the mechanical losses, such as bearing losses, and some of the hydraulic losses having been ignored in the simulation. However, the total error is less than 10%, which is within the allowable range. From the curve, the efficiency values for simulation and experiment are very close.

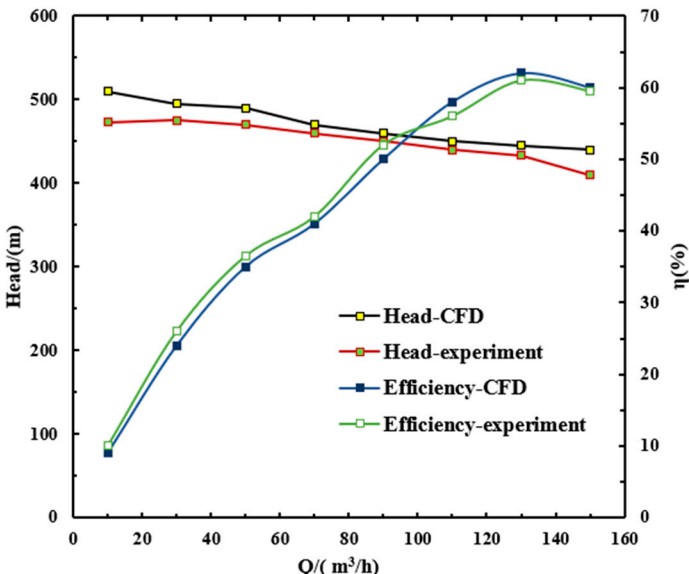

**Figure 4.** Pump performance curves at different flow rates.

Therefore, the above results showed that the numerical simulation method used in this paper is reasonable and can accurately predict the performance of the pump.

## 4. Numerical Setups

### 4.1. Pump Model

The high-speed centrifugal pump was modeled by Solidworks2019 software (Solid-Works, Welezi-Velakubray, France) in this study. The parameters of the pump are shown in Table 1. The detection points were set up at corresponding positions in the flow field as shown in Figure 5.

**Table 1.** Parameters of the high-speed centrifugal pump.

| Parameter | Value | Parameter | Value |
|---|---|---|---|
| Design flow rate $Q$ (m$^3$/h) | 130 | Speed $n_s$ | 9685 |
| Head $H$ (m) | 400 | Specific speed $n$ (r/min) | 76 |
| Number of impeller blades $Z_1$ | 6 | Inlet diameter of impeller $D_1$ (mm) | 98 |
| Outer diameter of impeller $D_2$ (mm) | 172 | Inlet width of impeller $b_1$ (mm) | 28.56 |
| Outer width of impeller $b_2$ (mm) | 10 | Blade inlet mounting angle $\beta_1$ (°) | 19 |
| Blade outlet mounting angle $\beta_2$ (°) | 37 | Initial diameter of short blade $D$ (mm) | 120 |
| Number of inducer blades $Z_2$ | 3 | Inducer lead $S_i$ (mm) | 52.5 |
| Tip diameter of inducer $D_t$ (mm) | 100 | Axial length of inducer blade $L$ (mm) | 70 |
| Leading edge angle $\theta_1$ (°) | 120 | Tip angle $\theta_2$ (°) | 360 |

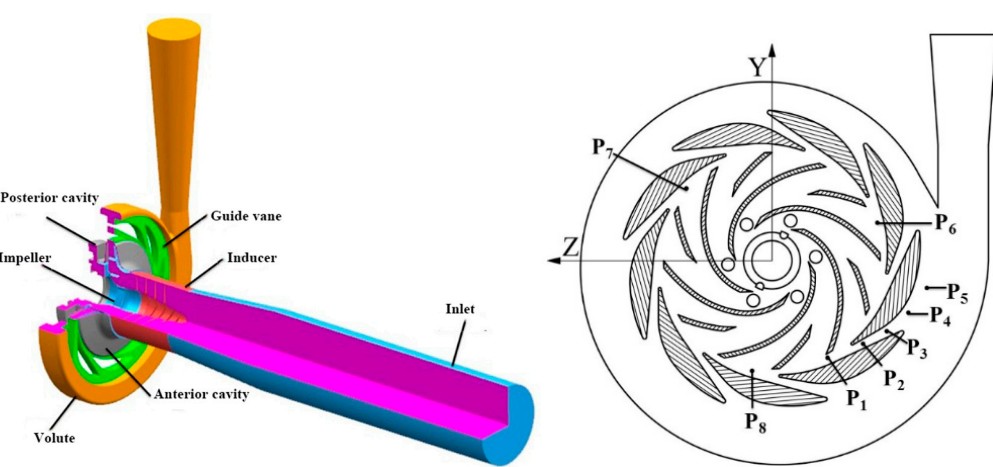

**Figure 5.** Pump model and detection points.

### 4.2. Mesh

The model was gridded by ICEM 18.0 software. The grid types of the computational models were mainly divided into structured and unstructured grids. The unstructured grid method was used in this study. The clearance of the impeller needed to be grid encryption. The grid is shown in Figure 6.

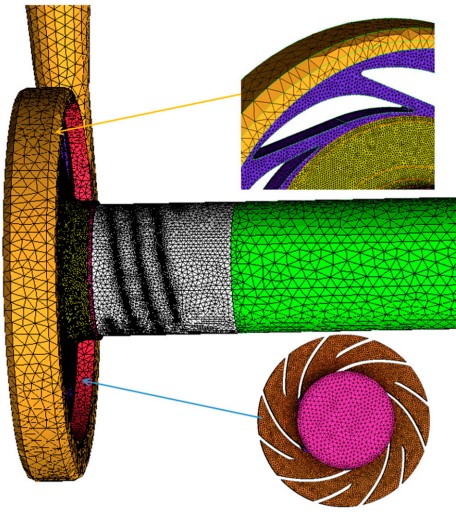

**Figure 6.** Computational domain grid schematic.

The grid independent analysis was carried out to eliminate the influence of the number of grid elements on the accuracy of the numerical simulation. The results are shown in Figure 7. It can be seen that the number of grid elements has a negligible effect on the hydraulic head when the number of grids is greater than 1.8 million. The grid wall function Y+ is less than 10 of the whole flow field, which satisfies the requirements of the turbulence model. In this study, 2.6 million grids were used.

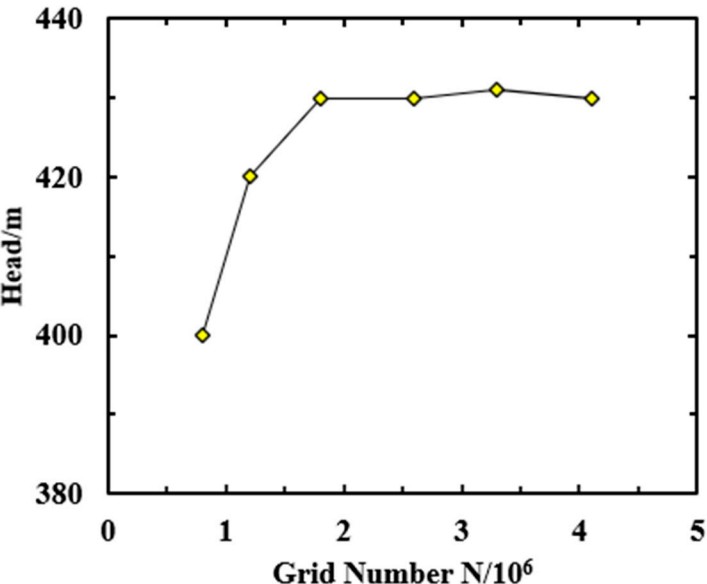

**Figure 7.** Analysis of mesh independence.

### 4.3. Turbulence Model

The numerical simulation was carried out using Fluent18.0, and the turbulence model used the SST k-ω model. The SST k-ω model includes all improved BSL k-ω models. These characteristics make it more accurate and reliable than the standard k-ω model and the BSL k-ω model. Previous BSL models combined the advantages of Wilcox and k-ε, but still failed to correctly predict the start and number of flow separations from smooth surfaces. The main reason is that the transport of the turbulent shear stress is not considered in either model. This leads to an overprediction of the eddy current viscosity. An appropriate transport formula can be obtained by limiting the eddy viscosity formula [14]:

$$\mu_t = \frac{\rho k}{\omega} \frac{1}{max\left[\frac{1}{\alpha^*}, \frac{SF_2}{a_1\omega}\right]} \tag{5}$$

In this equation, $S$ is the strain rate and $F_2$ is:

$$F_2 = tanh\left(\Phi_2^2\right) \tag{6}$$

$$\Phi_2 = max\left[2\frac{\sqrt{k}}{0.09\omega y}, \frac{500\mu}{\rho y^2\omega}\right] \tag{7}$$

where $y$ is the distance to the next surface.

### 4.4. Boundary Conditions

The boundary conditions were assumed to be mass flow rate at the inlet, and the outlet was set as the outflow. The non-slip boundary condition was applied to the wall of all flow regions. The calculated medium was water and its density is 998.1 kg/m³.

## 5. Numerical Results and Analysis

### 5.1. Pressure Analysis of Guide Vane

A graph of the pressure fluctuations at various monitoring points with different flow rates is shown in Figure 8. The pressure from P1–P5 increases gradually, according to the graph. The pressure of each point decreases gradually with the increase in the flow rate, especially the pressure at P2 and P3. It can be seen from the diagram that the pressure fluctuations at points P1–P5 have a periodicity, with six large peaks and six small peaks appearing in a cycle with different flow rates, which correspond to the six long blades and six short blades of the impeller. This is due to the static and dynamic interference between guide vanes. The pressure fluctuates when the blade sweeps the monitoring point. The pressure fluctuations of P1–P3 have more periodicity than those of P4 and P5. The pressure fluctuations at the monitoring points of P2 and P3 at a small flow rate are not as periodical as at a large flow rate.

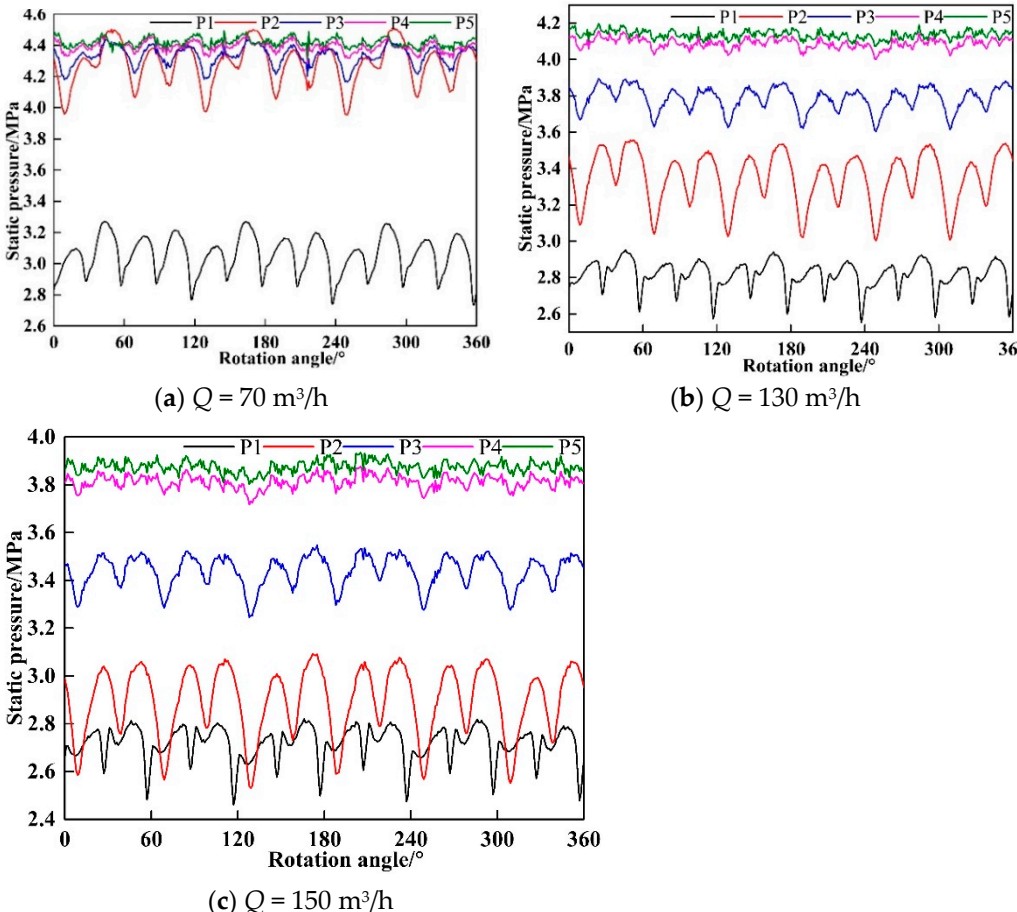

**(a)** $Q$ = 70 m³/h

**(b)** $Q$ = 130 m³/h

**(c)** $Q$ = 150 m³/h

**Figure 8.** The pressure fluctuations' value at each monitoring point at different flow rates in guide vane channel.

The time-domain diagram of the pressure fluctuations obtained by FFT transformation is shown in Figure 9. The design speed of the pump was 9685 rpm, so its shaft frequency was 161.7 Hz. The sweep frequencies of long blades and short blades were 1937 and 968.5 Hz, respectively, as the impeller has six long and six short blades. It can be seen from the diagram that the main frequency of pressure fluctuations is about 1937 Hz at P1, P2, and P3, and the secondary frequency is about 968.5 Hz, which correspond to 12 times and 6 times the shaft frequency, respectively. However, the primary and secondary frequencies at P4 and P5 deviate from the sweep frequencies of the long and short blades. Comparing the amplitude of the pressure fluctuations at different flow rates, it can be

found the amplitude of pressure fluctuations decreases at P1, increases at P2 and P3, but does not change significantly at P4 and P5.

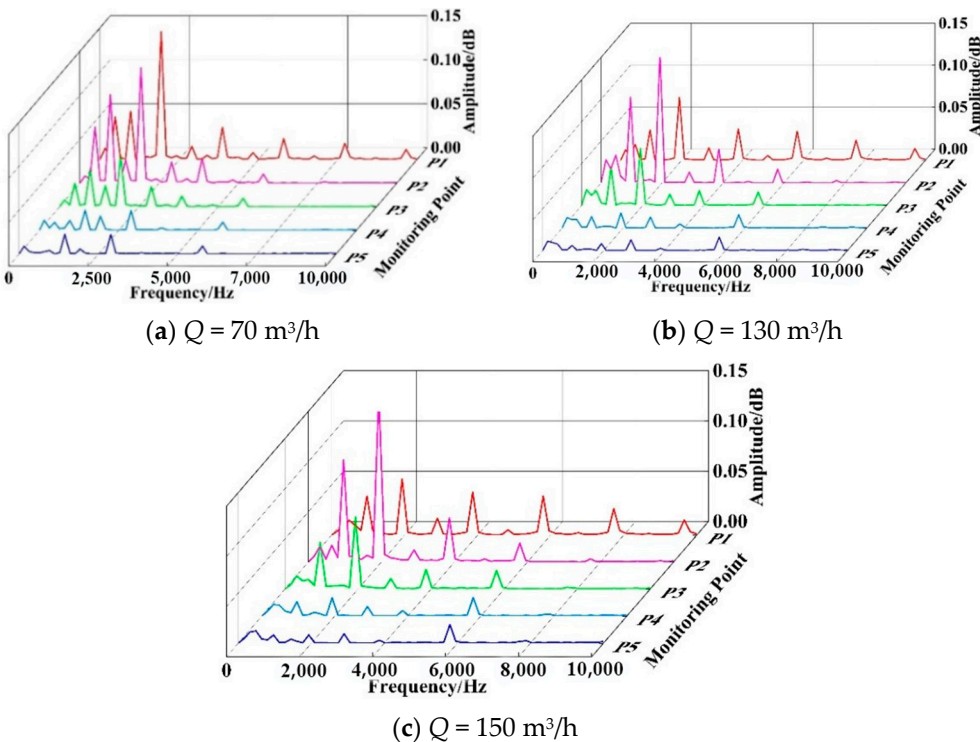

(**a**) $Q$ = 70 m³/h     (**b**) $Q$ = 130 m³/h

(**c**) $Q$ = 150 m³/h

**Figure 9.** Frequency domain diagram of pressure fluctuations at monitoring points in the guide vane flow channel.

The pressure distribution is closely related to the flow field distribution. The flow field with a time of $t_0$ was taken for analysis due to the periodicity of the pressure fluctuations, and the pressure streamline diagram is shown in Figure 10. The upper limit of the flow field pressure was 4 MPa to better reflect the pressure changes in the flow field. It can be seen that the pressure from the center of the impeller gradually increases outward. The pressure of P1–P5 increases as P1–P5 move away from the center of the impeller. As the flow rate increases gradually, the low-pressure area expands outward with a large flow rate. This is due to the larger the flow rate, the faster the pump flow, and the greater the pressure loss, which makes the low-pressure zone gradually expand to the guide vane channel. A large number of vortices appear in the impeller passage and guide vanes at design and at a small flow rate due to the backflow and secondary flow between the guide vanes and impellers, but fewer vortices at a large flow rate. This is due to the use of enlarged flow design. P4 and P5 are located at the exit of the guide vanes. The pressure fluctuations of the guide vanes do not change significantly at different flow rates, as the guide vanes play a bigger role in energy conversion. The pressures of P4 and P5 are more stable than those of P1–P3. This also shows that the guide vanes can play the role of collecting and diffusing the pressure. The design of the guide vane is reasonable and feasible.

### 5.2. Pressure Analysis of Static and Dynamic Interference

The impeller is the most important working part of a high-speed centrifugal pump. In this study, three monitoring points, P6, P7, and P8, were selected to measure the pressure in the interference zone between the impeller and the guide vanes, which corresponded to the three monitoring points in the experimental pump.

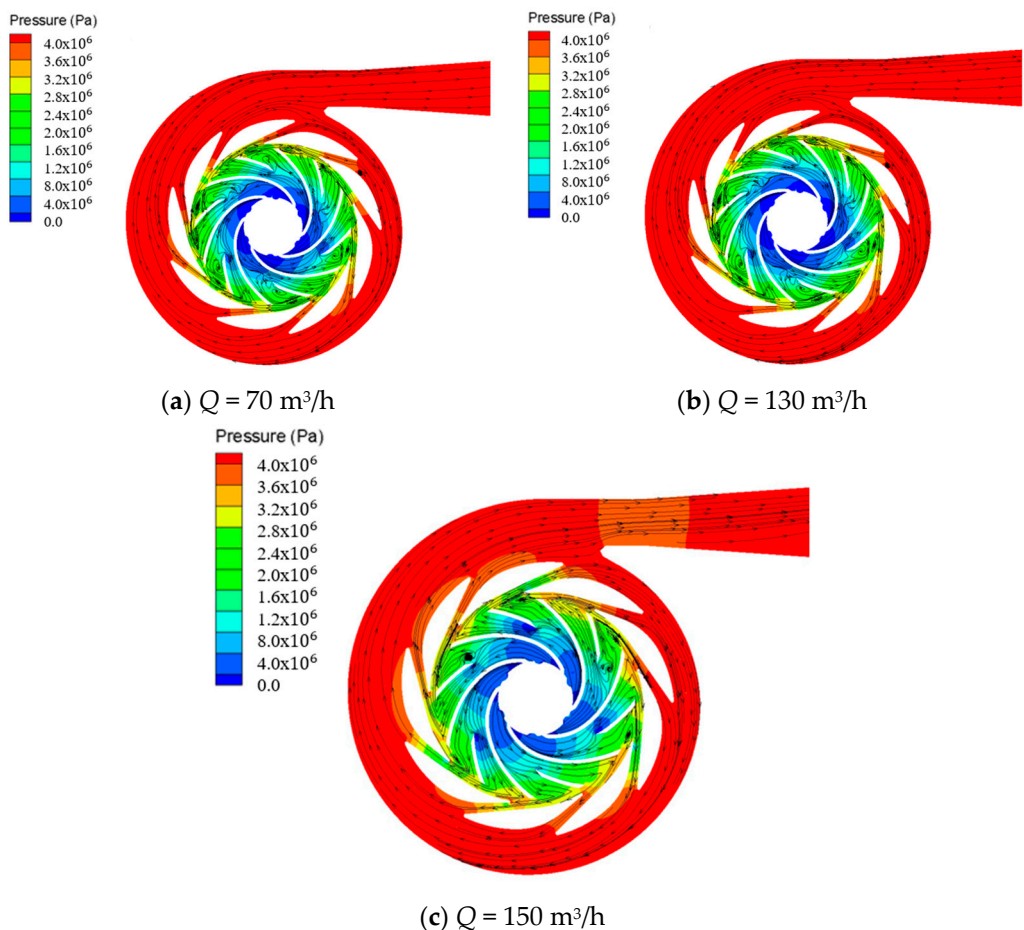

(**a**) *Q* = 70 m³/h         (**b**) *Q* = 130 m³/h

(**c**) *Q* = 150 m³/h

**Figure 10.** Figure of the pressure distribution and streamline.

Figure 11 shows the pressure fluctuations at the P6, P7, and P8 monitoring points under different flow conditions. It can be seen from the figure that the amplitude of pressure fluctuations at each monitoring point is large and that there is strong periodicity, which is related to the dynamic and static interference between the impeller and the guide vane. The static pressure at monitoring points P6, P7, and P8 increases gradually when the flow rate is 70 m³/h. When the flow rate is 130 m³/h, the static pressure at the monitoring point P6 is the largest, but the difference between P7 and P8 is not significant. Thus, with a change in the flow rate, the static pressure distribution at the static and dynamic interference also changes. The fluctuating pressure amplitude at the small flow rate of 70 m³/h is larger than that at the large flow rate of 150 m³/h. There are six large peaks and six small peaks in a cycle, which correspond to the six long blades and six short blades of the impeller. This shows that the pressure fluctuations at the static and dynamic interference is related to the number of impeller blades.

The data in Figure 12 were transformed by Fourier transform to obtain the frequency domain diagram of pressure fluctuations at P6, P7, and P8 under different flow conditions, as shown in Figure 12. It can be seen from the diagram that the primary and secondary frequencies of pressure fluctuations at P6, P7, and P8 are nearly identical, with a slight difference in amplitude. With an increase in the flow rate, the amplitude of the primary and secondary frequencies decrease gradually. In addition to the 12, 6, and 1 times shaft frequencies, the pressure fluctuations are affected by the interval of 12 times shaft frequency, which is caused by the superposition of the frequency.

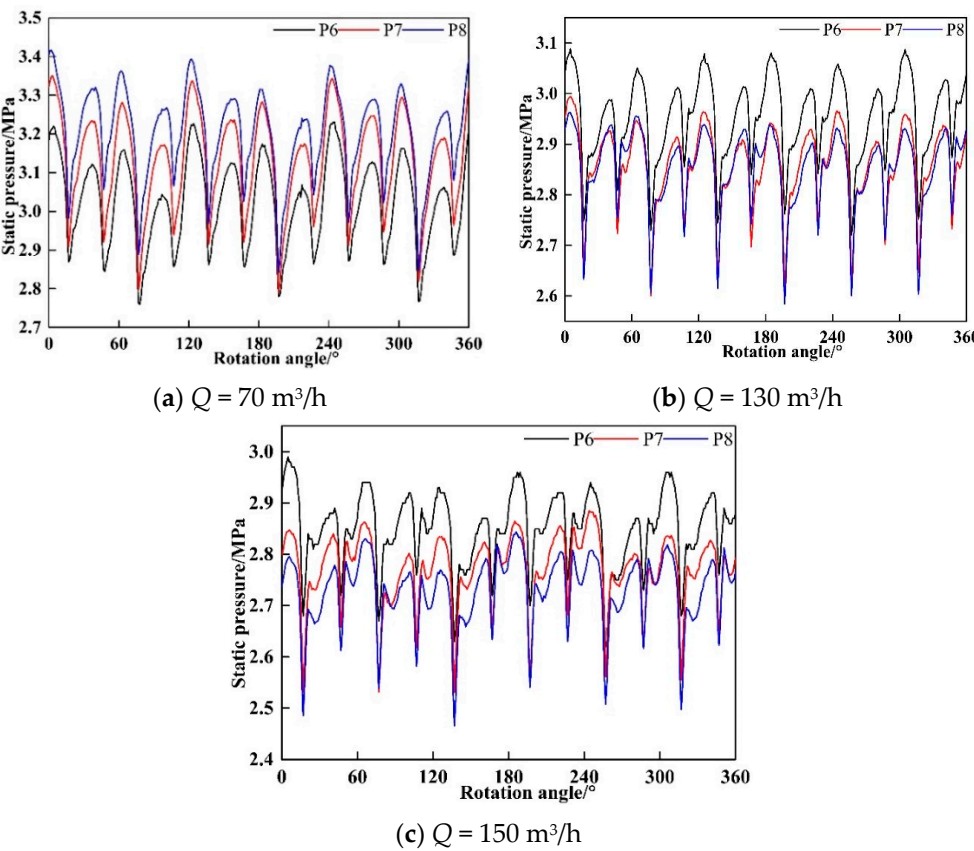

(**a**) $Q = 70$ m³/h      (**b**) $Q = 130$ m³/h

(**c**) $Q = 150$ m³/h

**Figure 11.** Pressure fluctuations at various monitoring points under different flow conditions.

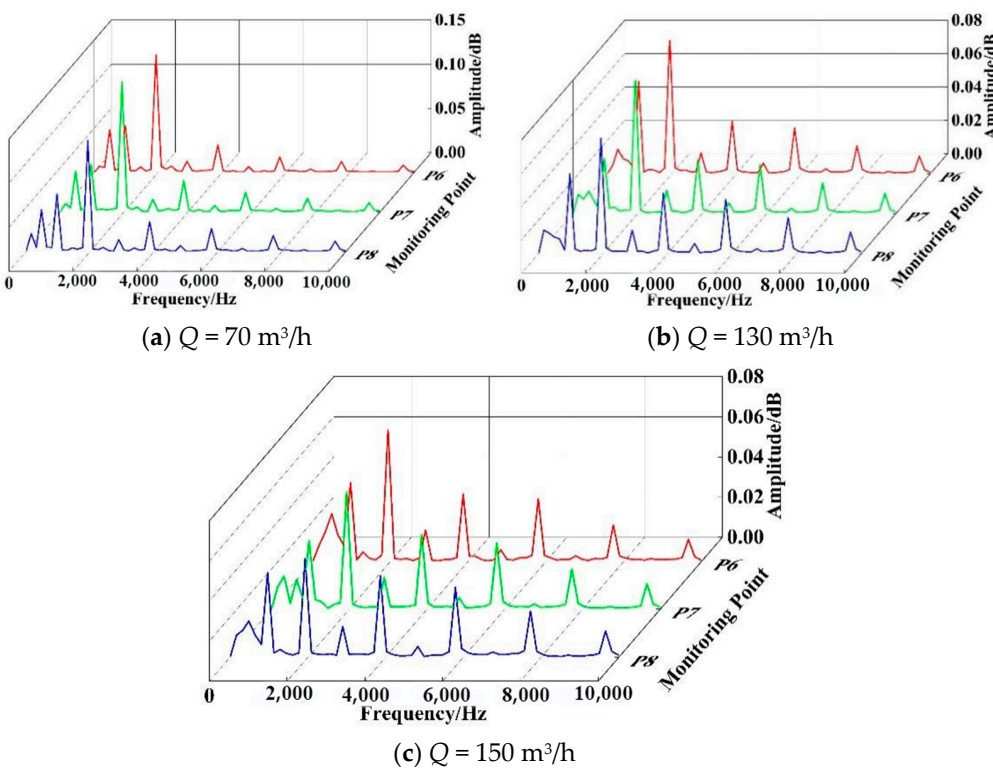

(**a**) $Q = 70$ m³/h      (**b**) $Q = 130$ m³/h

(**c**) $Q = 150$ m³/h

**Figure 12.** Frequency domain diagram of pressure fluctuations at monitoring points P6, P7, and P8.

### 5.3. Vorticity Analysis

A vorticity diagram can reflect the flow in a pump, and is often used to represent the curl characteristics of the flow field. The pressure fluctuations were analyzed by a vorticity diagram. The vorticity diagram at $t_0$ time was taken for analysis, as shown in the Figure 13. It can be seen from the vorticity diagram that the strong vortices are mainly concentrated near the velocity vorticity regions, such as the leading edge of the short blade, the outlet of the impeller, and the guide vane. The vortex at the leading edge of the short blade is caused by the flow separation and the flow obstruction of the short blade. As can be seen from Figure 13, the vortices at the leading edge of the short blade are weakest at 70 m$^3$/h. With an increase in the flow rate, the vorticity intensity increases for the flow separation at the leading edge of the short blade, becoming more drastic. The vortex at the impeller exit is caused by static and dynamic interference between the impeller and the guide vane. The vortex at the exit of the guide vane is caused by flow separation at the exit, which is different from the flow velocity direction in the volute. With an increase in the flow rate, the flow separation and vortex at the exit of the guide vane gradually increase. According to the diagram, the vortex is strongest at 150 m$^3$/h and weakest at 70 m$^3$/h.

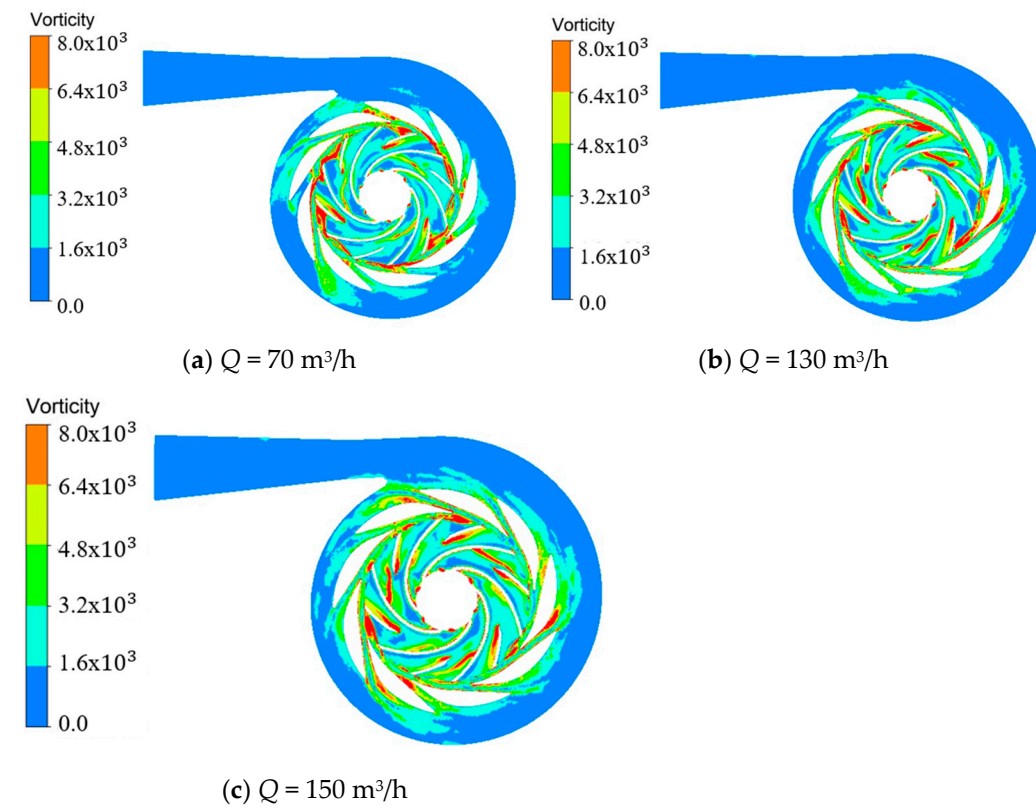

(**a**) $Q$ = 70 m³/h                            (**b**) $Q$ = 130 m³/h

(**c**) $Q$ = 150 m³/h

**Figure 13.** Vorticity distribution at different flow rates.

In order to further explain the relationship between vorticity and pressure fluctuations, the Q criterion was used to visualize the vorticity. The value of Q was assigned as 0.01, and speed was color-coded. A vorticity diagram over a period of time was taken for analysis. Strong vortices mainly concentrate at the outlet of the impeller and the guide vane, as shown in Figure 14. As for the vortex at the impeller outlet, the strength is stronger at a low flow rate of 70 m$^3$/h due to the backflow and secondary flow in the flow channel of the impeller. With an increase in the flow rate, the secondary flow and backflow in the impeller channel gradually weaken, along with the size and intensity of the vortex. The vortex strength at the impeller outlet is weak at design and at a large flow rate. Regarding the vortex at the exit of the guide vane, its strength is increases with the an increase in the flow rate. The strong vortex at the impeller outlet has a great influence on the pressure

fluctuations of the impeller passage and the guide vane inlet. The pressure amplitudes of P1 and P6–P8 near the impeller outlet are greatly affected. The influence of the vorticity of the impeller outlet on the pressure fluctuations amplitude is weakened when P2–P5 are distant from the impeller outlet area. The vortex at the impeller is closely related to the static and dynamic interference between the impeller and the guide vane, which makes the vortex appear periodically. Therefore, the pressure fluctuations at P1 and P6–P8 have strong periodicity. P2–P5 pressures pulse periodicity and gradually weaken. The vortex intensity at the impeller outlet is closely related to the backflow and secondary flow, which further weaken the periodicity of the pressure at P2 and P3 at a low flow rate. With an increase in the flow rate, the strong vortex at the impeller outlet diffuses to the guide vane channel gradually, and the pressure at P2 and P3 becomes periodic. The vortex shape changes little at different flow rates and at different times. At a low flow rate, the intensity of the vortices changes noticeably at different times, which makes the amplitude of the pressure fluctuations of P1 and P6–P8 stronger. Under a standard flow rate and a large flow rate, the intensity of the vortex is weakened, and the intensity change is not noticeable, so the pressure fluctuations is weakened. The change in the vortex at the exit of guide vane is less drastic than that at the exit of the impeller, which makes the pressure fluctuations weaker.

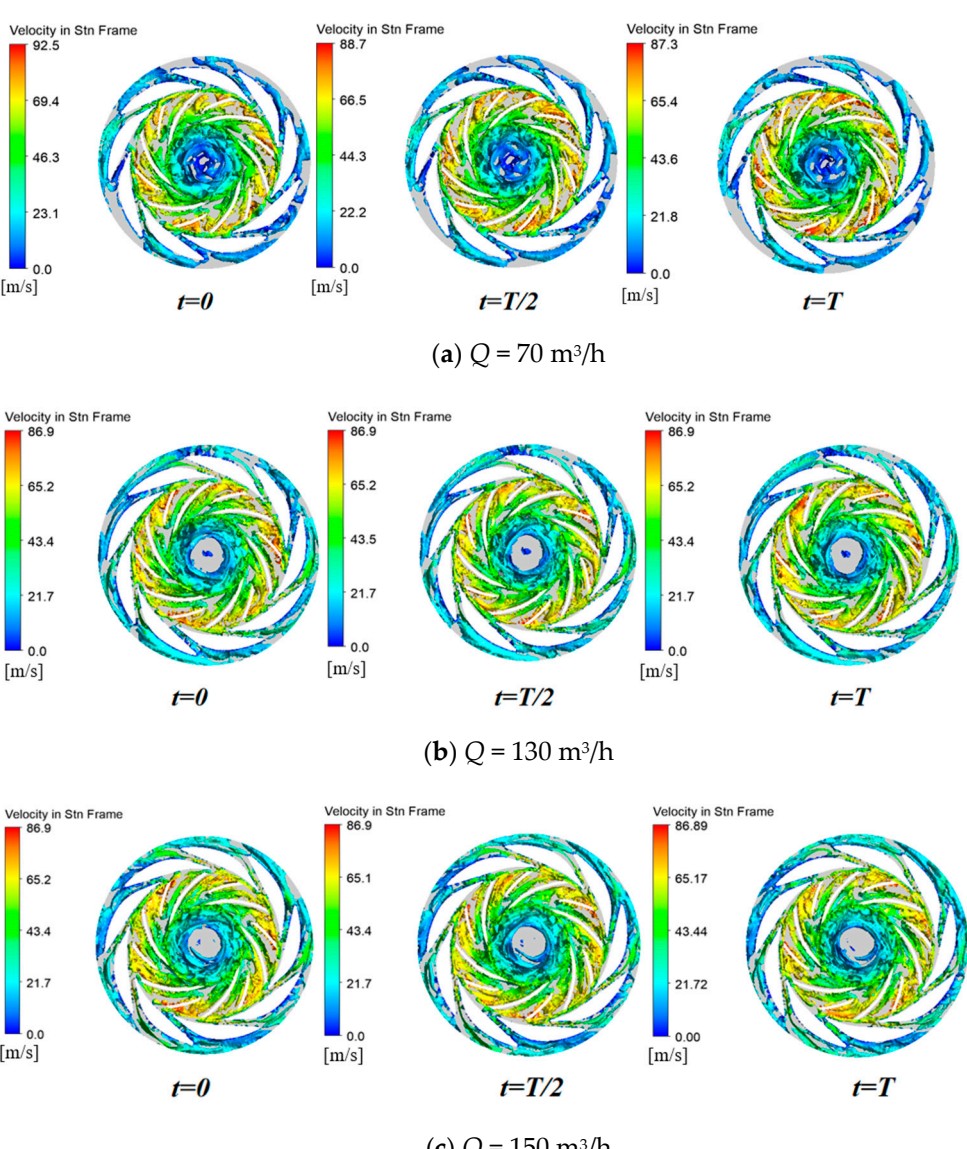

**Figure 14.** Vorticity distribution over a period of different flow rates.

### 5.4. Vibration Analysis

There are many factors affecting the vibration of a pump. This study showed that pump vibration is closely related to pressure fluctuations, and these pressure fluctuations can be analyzed according to the vibration. Figure 15 shows the vibration displacement diagram of the drive bearing and the non-drive bearing of the pump under different flow conditions. As can be seen from Figure 15, the vibration displacement at the non-drive bearing is significantly larger than that at the drive bearing. This is due to the fact that the axial displacement of the drive bearing was set to zero and the drive end was connected to the coupling during the operation. The impeller took 6.2 ms to complete one revolution. There were three peaks within 20 ms of the impeller revolving, which meant that the centrifugal pump shaft vibration was also periodic. There were 12 small peaks in every period, which indicated that the vibration was related to the shaft frequency and the impeller frequency. The vibration displacement of the centrifugal pump was the largest when the flow rate was 70 m$^3$/h, which corresponded to the pressure fluctuations.

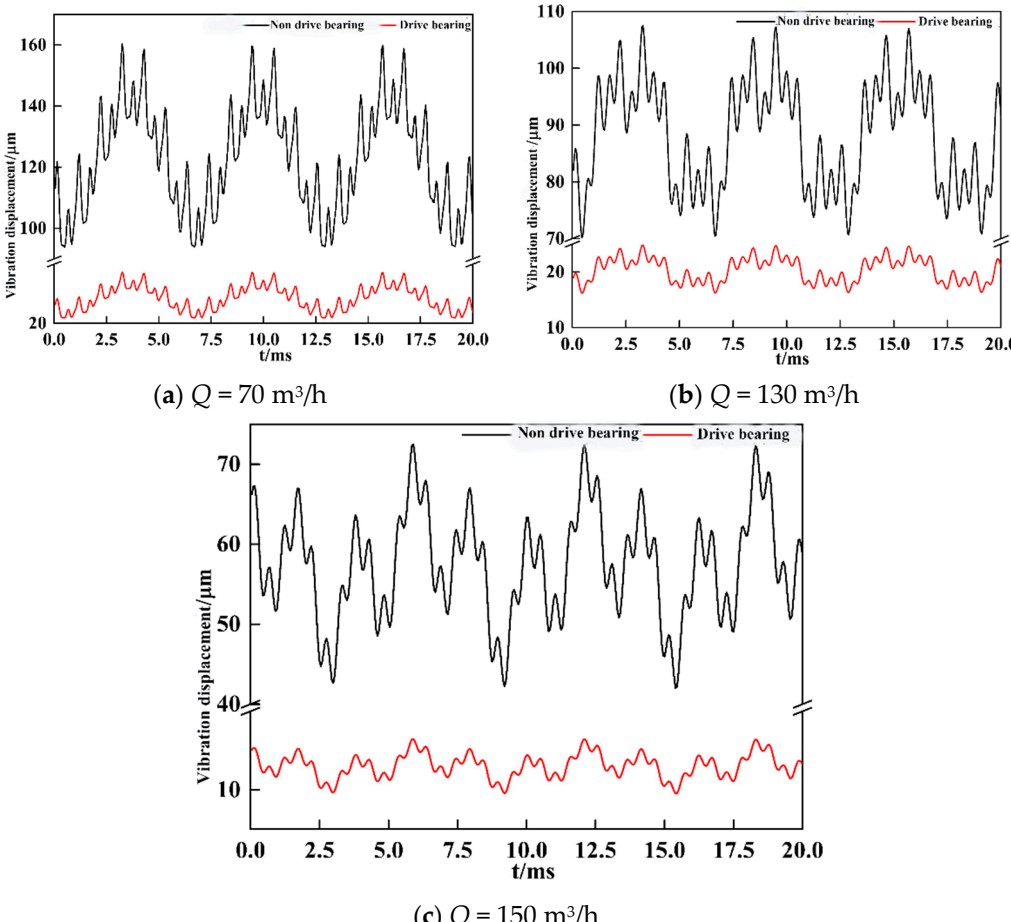

**Figure 15.** Vibration displacement at different flow rates.

### 6. Conclusions

The pressure fluctuations of a high-speed centrifugal pump with an enlarged flow design was studied in this paper, which can provide reference for pumps with an enlarged flow design. The conclusions are as follows:

(1) The pressure fluctuations at the impeller outlet and the guide vane passage are closely related to the change in the vortex at the impeller outlet. The vortices at the impeller outlet are mainly caused by the separation of fluid, and the vortices change periodically. The farther away from the exit of impeller, the weaker the fluctuations'

periodicity. With an increase in the flow rate, the pressure fluctuations in the guide vane become periodic.

(2)　The shape of the vortex of a pump with an enlarged design does not change significantly at different times. At a low flow rate, the change of intensity is significant and the amplitude of pressure fluctuations is large. The variation of the vortex is not significant under design and at a large flow rate, and the amplitude of the pressure fluctuations at a large flow rate is smaller than that under design conditions.

(3)　The flow field in the guide vane is stable, and the vortex at the outlet of the guide vane is stable at different flow rates. Therefore, the pressure at the outlet of guide vane is stable, which indicates that the guide vane can be added to suppress the pulsation for a pump with an enlarged flow design.

(4)　The vibration amplitude and the efficiency at a large flow rate are smaller than that under the designed condition. The rated pump capacity can be enlarged to restrain the pressure pulsation when the pump is designed by the enlarged flow method.

**Author Contributions:** Conceptualization, J.Z.; methodology, J.Z. and H.Y.; software, Y.L.; validation, H.Y., Y.L., H.L. and L.X.; formal analysis, H.Y.; investigation, L.X.; resources, J.Z.; data curation, J.Z.; writing—original draft preparation, H.Y.; writing—review and editing, H.Y.; visualization, H.Y., H.L. and L.X.; supervision, J.Z.; project administration, J.Z. and H.Y.; funding acquisition, J.Z. All authors have read and agreed to the published version of the manuscript.

**Funding:** This work was supported by the NSFC program (No. 52071296), and the key research and development program of Zhejiang Province (No. 2020C01027, and No. 2020C03099).

**Institutional Review Board Statement:** Not applicable.

**Informed Consent Statement:** Informed consent was obtained from all subjects involved in the study.

**Data Availability Statement:** The data presented in this study are available on request from the corresponding author. The data are not publicly available due to the policy of the Province Key Lab of Fluid Transmission Technology, Zhejiang Sci-Tech University, Hangzhou.

**Conflicts of Interest:** We confirm that no conflict of interest in this paper.

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
