# Peer review of "Pressure Fluctuation Characteristics of High-Speed Centrifugal Pump with Enlarged Flow Design"

_processes, doi:10.3390/pr9122261_

Round 1
Reviewer 1 Report
The article tackles a very interesting topic about operational instabilities within high-speed centrifugal pump with enlarged flow design. The introduction, applied methodology, analysis and research results have been well presented and expounded. The recapitulating parts of abstract and conclusions are concise and reflect the entire article content. Attained results support the article contribution to enriching the available knowledge of pumps towards an evidence-based solution. However, the following minor adjustments can be suggested for readership ease:
- The first letters of words in the title should be capitalized except conjunctions
- In the introductory background, Reference 2 (Luo) and reference 9 (Zhang) can be placed closely as they talk about dynamic and static interference between various pump components.
- As the pressure fluctuation/pulsation covers a wide frequency spectrum with varying amplitudes, it is better the keep the plural form, like fluctuations or pulsations
- Formatting for Figures and their titles should be centered
Fig 1: Revise the typo error for Pressure pulsation sensor
Fig. 2 As done for the test rig components, it would be better to show the position of visible sensors by allows
Line 8: ….under different structural designs and flow structures...; Line 12, 13: …flow rates...; Line 15: …results…; Line 23: …develops high…; Line 107: …sensor. Three…; line 186: …has more periodicity than…; line 331: there is a mention of Fig. 4.11, which does not correspond to the naming of figures in the article.
The conclusion for the model validation (lines 129, 130 and 131) should come at the end of the section (below Fig. 4).
Table 1 formatting should have hidden lines. Only the upper first, second and last lines
It would be better to include supporting references in the selection of the simulation model; as to where BSL failed to correctly predict the start and number of flow separations from smooth surfaces.
Author Response
Dear Reviewers:
Thank you for giving us the opportunity to submit a revised draft of the manuscript “Research on the pressure fluctuation characteristics of high-speed centrifugal pump with enlarged flow design, Manuscript ID: processes-1501013” for publication in the Journal of Processes. We are grateful for the insightful comments on our paper. According to your comment, we have revised the manuscript extensively. If there are any other modifications we could make, we would like to modify them and we really appreciate your help. And the responds to the comments are as following:
Point 1: The first letters of words in the title should be capitalized except conjunctions.
Response 1: We are really sorry for our careless. Thank you for your reminder.
Point 2: In the introductory background, Reference 2 (Luo) and reference 9 (Zhang) can be placed closely as they talk about dynamic and static interference between various pump components.
Response 2: We sincerely appreciate the valuable comments. We have checked the literature carefully and reference 2 (Luo) and reference 9 (Zhang) has been placed closely.
Point 3: As the pressure fluctuation/pulsation covers a wide frequency spectrum with varying amplitudes, it is better the keep the plural form, like fluctuations or pulsations.
Response 3: We feel sorry for our careless. In our resubmitted, the typo is revised. Thanks for your correction.
Point 4: Formatting for Figures and their titles should be centered
Fig 1: Revise the typo error for Pressure pulsation sensor
Fig. 2 As done for the test rig components, it would be better to show the position of visible sensors by allows
Line 8: ….under different structural designs and flow structures...; Line 12, 13: …flow rates...; Line 15: …results…; Line 23: …develops high…; Line 107: …sensor. Three…; line 186: …has more periodicity than…; line 331: there is a mention of Fig. 4.11, which does not correspond to the naming of figures in the article.
The conclusion for the model validation (lines 129, 130 and 131) should come at the end of the section (below Fig. 4).
Table 1 formatting should have hidden lines. Only the upper first, second and last lines
Response 4: Thanks for your careful checks. We have corrected the errors accordingly.
Point 5: It would be better to include supporting references in the selection of the simulation model; as to where BSL failed to correctly predict the start and number of flow separations from smooth surfaces.
Response 5: Thank you for pointing this out. This comes from the Fluent theory guide. We'll consider adding it to the references.
We feel thanks for your professional review work on our article. According to your suggestions, we have revised our manuscript. Thank you again for your comments and valuable suggestions.

Reviewer 2 Report
The work is relevant, scientifically grounded, but the issues raised by the authors have high scientific competition and a lot has already been researched in this area. Unfortunately, the authors do not say if the results of the work can be applied to other pump designs. Most of the results obtained in this work have already been partially sanctified by other authors:
1. Numerical Investigation on Vortex Pump, vorgelegt von M. Sc. Yang Song ORCID: 0000-0002-2018-6872, genehmigte Dissertation
2.https://doi.org/10.21595/jve.2017.18830
3.https://doi.org/10.1115/1.4051386
4. It would be more informative if the authors indicated the installation location of the monitoring points in Figure 2 or 3 (P6, P7, P8)
5. In lines 259, 279, 331 incorrect references to figures are given
The authors should emphasize the scientific novelty of the work done and their contribution to the research results.
It is desirable to give more detailed recommendations for the design of such pumps and further planned research in this area.
I hope this helps the authors of the article.
Author Response
Dear Reviewers:
Thank you for giving us the opportunity to submit a revised draft of the manuscript “Research on the pressure fluctuation characteristics of high-speed centrifugal pump with enlarged flow design, Manuscript ID: processes-1501013” for publication in the Journal of Processes. We are grateful for the insightful comments on our paper. According to your comment, we have revised the manuscript extensively. If there are any other modifications we could make, we would like to modify them and we really appreciate your help. And the responds to the comments are as following:
Point 1: The work is relevant, scientifically grounded, but the issues raised by the authors have high scientific competition and a lot has already been researched in this area. Unfortunately, the authors do not say if the results of the work can be applied to other pump designs. Most of the results obtained in this work have already been partially sanctified by other authors:
1. Numerical Investigation on Vortex Pump, vorgelegt von M. Sc. Yang Song ORCID: 0000-0002-2018-6872, genehmigte Dissertation
2.https://doi.org/10.21595/jve.2017.18830
3.https://doi.org/10.1115/1.4051386
Response 1: Thank you for pointing this out. The conclusions of this paper can be applied to the pump with enlarged flow design. We are sorry for not specifying it in the article. This article first analyzes the pressure pulsation of the high-speed centrifugal pump of the enlarged design. From its analysis result, there is indeed a lot of places similar to ordinary pumps, but there are different places. This article is intended to explore this kind of pumps. The result is helpful for the design of the pump with enlarged design such as working conditions, pressure pulsation suppression. And we will improve this in the conclusions of the paper.
Point 2: It would be more informative if the authors indicated the installation location of the monitoring points in Figure 2 or 3 (P6, P7, P8)
Response 2: We are really sorry for my careless. Thank you for your reminder.
Point 3: In lines 259, 279, 331 incorrect references to figures are given.
Response 3: We feel sorry for our careless. In our resubmitted, the typo is revised. Thanks for your correction.
Point 4: The authors should emphasize the scientific novelty of the work done and their contribution to the research results.
It is desirable to give more detailed recommendations for the design of such pumps and further planned research in this area.
Response 4: We think this is an excellent suggestion and we have revised the introduction and conclusion in this manuscript, which emphasize the novelty the work. The conclusions give more detailed for the design of such pumps.
We feel thanks for your professional review work on our article. According to your suggestions, we have revised our manuscript. Thank you again for your comments and valuable suggestions.
